# Age-Related Changes in Accuracy and Speed of Lateral Crossing Motion: Focus on Stepping from Leaning Position

**DOI:** 10.3390/ijerph19159056

**Published:** 2022-07-25

**Authors:** Yusuke Maeda, Daisuke Sudo, Daiki Shimotori

**Affiliations:** 1Department of Physical Therapy, School of Health Sciences at Odawara, International University of Health and Welfare, Yokosuka 250-8588, Japan; d.sudo@iuhw.ac.jp; 2Komaki City Hospital, Komaki 485-8520, Japan; shimo.daiki.040127@gmail.com

**Keywords:** motion accuracy, elderly, motion capture, obstacle crossing, fall prevention

## Abstract

Fall incidents are increasing every year and prevention is necessary. Preventing falls can increase the quality of life of the elderly and decrease medical costs. Stumbling and tripping are the main causes of falls and falls in the lateral direction, causing the hip fracture. This study aimed to analyze the accuracy and speed of lateral obstacle crossing in the elderly, especially from leaning posture. Twenty healthy older adults (6 men and 14 women, aged 71.7 ± 1.5 years) and 20 healthy young adults (5 men and 15 women, aged 21.4 ± 1.2 years) participated in this study. We set four conditions (normal, fast, leaning, and leaning fast), and participants crossed the obstacle laterally ten times under each condition. The crossing motion was captured using a three-dimensional analysis system. The trajectory of the foot, landed position, step time, center of gravity of the body, and moment of the lower extremity during the swing phase were calculated and compared between older and younger adults. In the leaning condition, the step time and knee moment of the elderly were significantly longer and larger than those of young adults. From the results of the trajectory of the foot and landed position in the leaning condition, motion inconsistency of the foot was found in the elderly. We believe that it is difficult for the elderly to perform the intended crossing motion and swing quickly because of aging. This inconsistency in motion is a serious cause of falls in the elderly.

## 1. Introduction

The number of fall accidents is increasing annually in many countries owing to the growing aging population. Patients are forced to undergo long-term medical treatment, which induces physical and mental hypofunction. Their daily life activities decrease, and medical and care costs increase. Therefore, preventing falls is important from a medical and social perspective. The major reasons for falls are stumbling and tripping. When these occur in the elderly, they need to take rapid and accurate steps to avoid falls. To achieve this, proprioception and vision need to work properly, and the muscles should work sufficiently. However, sensory and motor functions in the elderly are in decline. Henry et al. reported that the leg proprioception is changed with age [1]. Franz et al. reported that older adults tend to rely on visual sensations instead of declining proprioception [2]. Deshpande et al. reported that a decline in the proprioceptive system with age causes poor postural control [3]. Vision is essential to feedforward the motor command of the required height for a step, as well as the relative positional relationship between the body and obstacle [4]. However, some elderly individuals have impaired visual acuity and unstable postural control [5]. Galna et al. reported that older adults reduce the moment of the lower extremities in step [6]. It is necessary to clarify the risk factors related to this kind of sensory decline in the elderly. To avoid falls, it is essential to investigate not only anterior-posterior (A-P) balance ability, but also lateral balance ability. A previous report pointed out the risk of lateral falls, which are more likely to cause hip fracture than the A-P falls [7,8]. In terms of lateral balance, there are some differences between the elderly and young adults. Maki et al. reported that the elderly tend to take a step laterally using an unloaded leg (crossover step) [8]. For example, when the body is pushed rightward by a perturbation from the left, they would cross their left leg to the right, not using the right leg. Conversely, young adults use their right leg to step laterally (loaded side step) in the same situation. This means that it is difficult for the elderly to lift off the leg with a loaded weight. This strategy is unsafe because the crossing leg is likely to collide with the other leg. Therefore, it is important for the elderly to use a loaded side step to regain an upright stance. When it is an imminent fall, accurate and immediate stepping is needed to prevent the fall. However, the elderly cannot do so against the unpredictable perturbations [9,10]. We found that the step motion of the elderly lacks accuracy [11]. The foot trajectory during step motion was not consistent for the forward step. Similarly, the foot clearance of the gait of the elderly is more variable than that of young adults [12,13,14]. Therefore, it is necessary to investigate the accuracy of the lateral step motion and motion speed. Michalska et al. reported that the transit time during step initiation in older adults extended compared with that in the younger population [15]. This means that it might be difficult for older adults to control their center of gravity (COG) and shift it quickly for stepping. Therefore, we focused on the lateral stepping motion of the elderly, especially in the lateral leaning position, because it is important for older adults to step quickly and accurately by the loaded side step to avoid falls. We aimed to investigate the accuracy and velocity of lateral stepping motion of the elderly from both the normal stance and the lateral leaning position. 

## 2. Materials and Methods

### 2.1. Participants

Twenty healthy older adults (6 men and 14 women; aged 71.7 ± 1.5 years; height 153.5 ± 4.2 cm; weight 51.6 ± 6.8 kg) and 20 healthy young adults (5 men and 15 women; aged 21.4 ± 1.2 years; height 164.4 ± 7.3 cm; weight 56.2 ± 7.6 kg) without any neurological and musculoskeletal diseases were enrolled (Table 1). The participants were given a written explanation of the experimental purpose, procedures, potential risks, and the right to refuse the cooperation for the study. All participants provided written informed consent to participate in this study. The experimental procedures were approved by the ethics committee of the International University of Health and Welfare. This study was performed in line with the principles of the Declaration of Helsinki. We recruited participants from a Silver Human Resource Center for the elderly. We also recruited young participants using a bulletin board on campus.

### 2.2. Experimental Setup

The crossing motion was captured using a three-dimensional analysis system (Vicon Motion System Ltd., Oxford, UK). Nine cameras and four force plates (AMTI Inc., Watertown, MA, USA) were synchronized. The sampling rates of the cameras and force plates were 100 Hz and 1000 Hz, respectively. We defined Y as the anterior direction (anterior, +), X as the lateral direction (right, +), and Z as the vertical direction (upward, +). Reflective markers of 14 mm diameter were placed according to the Plug-in Gait marker set following 39 locations: the left and right skull of the frontal head, left and right skull of the rear head, 7th cervical spinous process, 10th thoracic spinous process, suprasternal notch, sternal xiphoid, left and right acromioclavicular joint, lateral epicondyle of the left and right humerus, left and right radial styloid processes, left and right ulnar styloid processes, left and right second metacarpal head, left and right center of humerus, left and right center of ulna, left and right anterior superior iliac spine, left and right posterior superior iliac spine, left and right center of lateral thigh, left and right center of lateral lower thigh, medial joint space in left and right knee, center of the lateral malleolus of the left and right ankles, head, left and right second metatarsal head, left and right calcaneus, and right scapula.

### 2.3. Procedure

We asked the participants to cross over the obstacle, which was 15% of their height. The dominant foot was used for the crossing. The dominant foot was defined as the foot used for kicking a ball. The obstacle was placed 10 cm lateral to the participants’ right foot. They crossed the obstacle with their right leg from upright standing and then stopped the motion after landing (Figure 1). They were asked not to look at their feet during the motion. We also asked them to practice this motion several times as a preparatory exercise and confirmed whether they could understand our instructions. We set four conditions to analyze the crossing motion: (1) Normal condition: crossing over the obstacle from upright stance position with normal and natural speed; (2) Fast condition: crossing over the obstacle from the upright stance position as fast as possible; (3) Leaning condition: crossing over the obstacle from a lateral leaning posture with normal and natural speed; (4) Leaning fast condition: crossing over the obstacle from the lateral leaning posture as fast as possible.

The participants crossed the obstacle ten times in each condition. Each participant performed a total of 40 times (ten trials ×4 conditions). They took a 5-min rest for every 10 times of the trial. To set the lateral leaning position, we measured the limit of stability using the force plate of each participant in advance. They attempted to tilt their body right from an upright standing posture to their maximum possible extended position without moving their feet. The maximum leaning position of the center of foot pressure (COP) was defined as 100%. In the leaning and leaning fast condition, participants waited at a 70% lateral leaning position and then crossed over the obstacle at the examiner’s cue. The examiner monitored the COP position in real-time to maintain the participants in the 70% leaning position for 1–2 s, and then gave the cue for crossing motion. To evaluate the motion consistency, we asked the participants to perform the same motion ten times; they were instructed to swing their leg to follow the same trajectory in each trial and land their foot at the same position as much as possible. If the participant tripped or stumbled on the obstacle, the trial was excluded.

### 2.4. Data Analysis

The motion analysis software Visual 3D (C-Motion, Inc., Washington, DC, USA) was used to analyze the motion capture data. The coordinates of the reflective markers during the swing phase were extracted. The start and end points of the data were defined as the moment when the heel marker moved 2 mm (heel off) and the force plate detected 10 N of the foot contact (landing). Each data point was normalized from zero to one. To calculate the average trajectory of ten data, the curves were approximated using a cubic spline function, and the data were resampled at regular intervals. The gap distance between the average trajectory and each trial were summed and defined as the error distance. A high value of the error distance means that the trajectories of the trials are not consistent. We used the same procedure as in a previous study [11,16]. 

The COG displacement (m), moment of the left hip, left knee, left ankle (Nm/kg), standard deviation (SD) of the right heel marker, step time (s), and error distance (m) were obtained. The maximum COG displacement during the swing phase in each direction (X, Y, and Z) was calculated. The moment of the hip (extension), knee (extension), and ankle (plantar flexion) were also the maximum values during the crossing motion. The SD of the right heel marker of ten trials in the horizontal plane was calculated using the coordinates of the landed position for the X (SD_X) and Y (SD_Y) directions. A high value of the SD indicates inconsistent landed positions. The tenth trial of each condition was adopted for the calculation of all parameters, except for the foot trajectory and landed position.

We used the Bayesian model to estimate each parameter of the posterior distribution. The Hamiltonian Monte Carlo (HMC) method, which is a type of Markov Chain Monte Carlo (MCMC), was used to calculate the posterior distribution. All parameters were assumed to have a Gaussian distribution, and a noninformative prior distribution was used. We applied four steps to obtain the posterior distribution as follows: iteration 2000, burn-in 500, thin 1, and chain, 4, thus 6000 random samples were obtained. In the case of Rhat ($\hat{R}$) < 1.05, we recognized that the posterior distribution converged. We calculated the posterior distribution from the mean value and obtained the Bayesian 95% credible interval (CI) of the mean. Moreover, the distribution of the differences in the mean values of each group was calculated. If the 95% CI of the distribution of differences did not include zero, we judged statistical significance.

We used R software, version 4.0.2 (The R Foundation for Statistical Computing, Vienna, Austria) and the statistical modeling computation Stan (Stan Development Team) for Bayesian analysis.

## 3. Results

$\hat{R}$ of all parameters were less than 1.05. Table 2 indicates the results and statistical information of all parameters. The COG displacement in the Z direction in the leaning fast condition of older adults was larger than that of young adults. The knee joint moment in the leaning condition of older adults was larger than that of young adults. SD_X and SD_Y in the leaning condition of older adults were larger than those of young adults. The error distance in the leaning condition of older adults was larger than that of young adults. The step time in leaning and leaning fast condition of the older adults were larger than that of the young adults. 

## 4. Discussion

The results of this study revealed that the accuracy of the crossing motion of the elderly under the leaning condition was inconsistent. The high values of SD_X and SD_Y indicated that the landed position did not match the targeted location in each trial. These results showed that the step motion was not controlled. Kulkarni et al. demonstrated that foot placement during gait was controlled to adjust the COG. However, this adjustment was weaker in older adults [17]. Similarly, the high values of error distance in the leaning condition indicate that the foot trajectory of the swing went through a different path each time. This inconsistency induces tripping or stumbling during stepping, especially in the leaning posture. We believe that this inconsistency might result from changes in muscle contraction. The elderly have increased coactivation of the lower muscles compared with young adults [1]. Yao et al. suggested that the elderly showed difficulties in eccentric contractions compared with young adults [18]. Moreover, the high values of knee extension moment of the leaning condition mean that the elderly exerted a larger effort for the crossing motion compared with young adults. We believe that the elderly have less capacity to regain stability in a leaning posture. Similarly, high values of knee extension moment might be related to the COG movement. The COG displacement in the vertical direction (Z) in the leaning fast condition of the elderly was larger than that of young adults. The same tendency was observed in the other three conditions; the elderly might lift their COG for clearance between the foot and the obstacle. However, there was no statistically significant difference in the normal, fast, and leaning conditions; therefore, further research is required. Vertical COG movement might be related to the step time. The step time of the elderly in the leaning and leaning fast conditions were longer than those of young adults. It is possible that the step time was extended because of the vertical motion or the lack of rapid motion. Rapid motion requires a large inertial force; therefore, the body tends to be unstable because of the force. A previous study reported that the elderly took additional steps in the lateral stepping reaction because the initial step lacked stability [8]. They pointed out that this reflected the decline in sensory function, fear of falling, and the loss of balance confidence. There are many reports of sensory dysfunction in the elderly population. The dynamic sensitivity decreases because of the morphological changes in muscle spindles in rats [19]. The nerve conduction velocity decreases because of smaller axons in cats [20]. Moreover, the decline in the brain, which is related to sensorimotor regions, induces poorer proprioceptive integration [21]. These changes in the sensory system, including signal transmission and integration into the central nervous system, may reflect in the motor skills. Thus, even if there were adequate sensory inputs, the elderly might not be able to utilize these. Klass et al. reported that the elderly rely more on central than peripheral mechanisms to adjust their ankle motion [22]. Therefore, a lack of sensory inputs or non-usage of sensory information potentially induces fear of falling or lack of confidence. 

This study had several limitations. We did not gather data on basic body function (such as muscle strength and sensory function) and balancing abilities. If we collected these data, we could have been able to identify the reasons for motion inconsistency. There were statistical differences between the elderly and young adults for SD_X, SD_Y, knee moment, and error distance in the leaning condition but not in the leaning fast condition. We could not explain this point in our study. We have no electromyogram (EMG) data; therefore, we cannot directly refer to muscular activity. Further investigations using motion capture and EMG are required.

## 5. Conclusions

The purpose of this study was to clarify the mechanism of the lateral crossing motion in the elderly using motion capture. Twenty healthy older and 20 healthy young adults performed lateral obstacle crossing ten times in four conditions as follows: normal, fast, leaning, and leaning fast. Based on Bayesian estimation, the step time and knee moment of the elderly in the leaning condition were significantly longer and larger than those of young adults. From the results of the trajectory of the foot and landed position in the leaning condition, motion inconsistency of the foot was found in the elderly. Moreover, postural instability was found in the leaning and leaning fast conditions. We believe that it is difficult for the elderly to cross the lateral obstacle as intended because of sensory deterioration and muscle dysfunction. These kinds of inconsistencies would induce falls, so it is necessary to further investigate the reasons.

## Figures and Tables

**Figure 1 ijerph-19-09056-f001:**
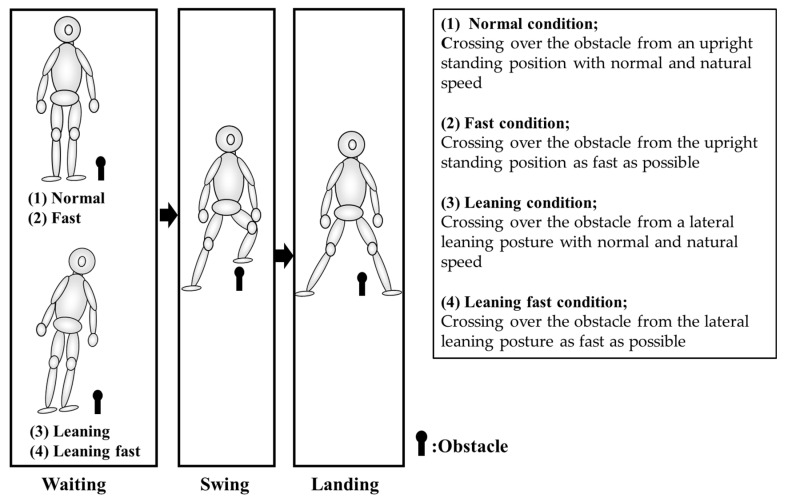
Crossing motion.

**Table 1 ijerph-19-09056-t001:** Participant’s characteristics.

	Older Adults (*n* = 20)	Young Adults (*n* = 20)
Age (years)	71.7 ± 1.5	21.4 ± 1.2
Height (cm)	153.5 ± 4.2	164.4 ± 7.3
Weight (kg)	51.6 ± 6.8	56.2 ± 7.6
Body mass index	21.9 ± 6.8	20.5 ± 1.4
The number of right-footed	20	20
The number of men	6	5

**Table 2 ijerph-19-09056-t002:** The results of mean, standard deviation, 95% CI of the distribution of differences, and Rhat.

	Older Adults		Young Adults		CI (2.5~97.5%)	Rhat	Statistical Significance
	Mean	SD	Mean	SD			
COG_X (m)							
Normal condition	0.27	0.14	0.25	0.03	−0.09~0.06	1	
Fast condition	0.34	0.30	0.23	0.04	−0.25~0.04	1	
Leaning condition	0.36	0.37	0.22	0.06	−0.32~0.04	1	
Leaning fast condition	0.40	0.42	0.21	0.04	−0.39~0.02	1	
COG_Y (m)							
Normal condition	0.06	0.09	0.03	0.01	−0.08~0.01	1	
Fast condition	0.08	0.08	0.04	0.02	−0.07~−0.01	1	
Leaning condition	0.07	0.08	0.04	0.02	−0.07~0.02	1	
Leaning fast condition	0.07	0.09	0.04	0.02	−0.08~0.01	1	
COG_Z (m)							
Normal condition	0.19	0.33	0.08	0.01	−0.28~0.06	1	
Fast condition	0.30	0.51	0.08	0.01	−0.47~0.03	1	
Leaning condition	0.31	0.55	0.08	0.02	−0.49~0.03	1	
Leaning fast condition	0.32	0.46	0.08	0.01	−0.48~−0.02	1	*
Hip Extension Momment (Nm/kg)							
Normal condition	0.27	0.14	0.25	0.03	−0.62~0.10	1	
Fast condition	0.34	0.30	0.23	0.04	−0.58~0.12	1	
Leaning condition	0.36	0.37	0.22	0.06	−0.76~0.22	1	
Leaning fast condition	0.40	0.42	0.21	0.04	−0.79~0.08	1	
Knee Extension Momment (Nm/kg)							
Normal condition	0.06	0.09	0.03	0.01	−0.33~0.02	1	
Fast condition	0.08	0.08	0.04	0.02	−0.29~0.02	1	
Leaning condition	0.07	0.08	0.04	0.02	−0.35~−0.05	1	*
Leaning fast condition	0.07	0.09	0.04	0.02	−0.28~0.13	1	
Ankle Plantar-flexion Momment (Nm/kg)							
Normal condition	0.19	0.33	0.08	0.01	−0.20~0.23	1	
Fast condition	0.30	0.51	0.08	0.01	−0.21~0.17	1	
Leaning condition	0.31	0.55	0.08	0.02	−0.14~0.42	1	
Leaning fast condition	0.32	0.46	0.08	0.01	−0.20~0.23	1	
SD_X							
Normal condition	0.03	0.03	0.02	0.01	−0.02~0.01	1	
Fast condition	0.02	0.03	0.02	0.01	−0.02~0.01	1	
Leaning condition	0.03	0.03	0.02	0.00	−0.03~0.00	1	*
Leaning fast condition	0.02	0.02	0.02	0.02	−0.01~0.02	1	
SD_Y							
Normal condition	0.02	0.02	0.02	0.01	−0.02~0.01	1	
Fast condition	0.02	0.02	0.01	0.01	−0.02~0.01	1	
Leaning condition	0.02	0.02	0.01	0.00	−0.02~0.00	1	*
Leaning fast condition	0.02	0.01	0.02	0.01	−0.02~0.01	1	
Error distance (m)							
Normal condition	0.018	0.008	0.017	0.005	−0.06~0.03	1	
Fast condition	0.016	0.007	0.016	0.005	−0.04~0.04	1	
Leaning condition	0.021	0.008	0.017	0.005	−0.08~0.00	1	*
Leaning fast condition	0.016	0.008	0.014	0.005	−0.07~0.02	1	
Step-time (s)							
Normal condition	0.83	0.09	0.81	0.09	−0.08~0.05	1	
Fast condition	0.66	0.08	0.62	0.05	−0.08~0.01	1	
Leaning condition	0.78	0.10	0.69	0.08	−0.16~−0.03	1	*
Leaning fast condition	0.63	0.08	0.54	0.07	−0.15~−0.04	1	*

CI: Credible Interval; *: Statistically significant.

## Data Availability

The datasets analyzed during this study are available from the corresponding author upon reasonable request.

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
