# Peer review of "Age-Related Changes in Accuracy and Speed of Lateral Crossing Motion: Focus on Stepping from Leaning Position"

_ijerph, 2022, doi:10.3390/ijerph19159056_

Round 1

Reviewer 1 Report

The title can be strengthened with the insertion of the methodology used.

Introduction: Some information lacks references; the objective, which may be more originating. The definition of the study objective should be enriched; The statement "We aimed to investigate the stepping of the elderly and clarify the differences between older and younger adults in terms of kinematics and kinetics" is very poor as a justification for the study, and does not report the importance that the study may have in society.

Discussion: Some statements lack bibliographic references. This chapter looks poor to me. I recommend that it be more in-depth, and there may be a greater connection between the results found and their consequences for this population.

Limitations and recommendations for future research can also be further explored.

References could be updated? There is only one with less than 5 years.

Author Response

Please confirm the uploaeda file.

Reviewer 2 Report

Taking rapid and accurate steps to avoid falls are very important in elders. It is a useful work to report differences in neuromuscular performance between young and elder. There are several details can be modified.

1)Whether the subjects are declared to have no disease which will affect the experiment?

2)The procedure of the test was not very clear. For example, before the test, the practice to prepared for all the procedure is needed especially in elders, and before the test, warm up procedure is also needed.

3) ‘The dominant foot was defined as the foot used for kicking a ball. All subjects’ dominant feet were on the right side.’ This maybe described in the 2.1. Participants as an entry criteria.

4) Four conditions are designed in this paper, the test sequence for the four conditions is not clear, random sequence can be more meaningful.

5) The rest between different conditions is not clear, how to maintain consistent physical fitness of the elders in different conditions is very important in the paper.

6)101-102, ’ Fast condition: crossing over the obstacle from the upright stance position as fast as possible;’ ‘as fast as possible ‘ cannot be quantified, so it seems not rigorous.

7)Is it possible to normalize the result according to individual gender, height, weight or other? If not, the difficulty should be discussed.

8)Four conditions represent different aspects of motion performance, while only motions in LEAN situation show difference between young and elder. Can it show that movement adjustment ability is similar between elder and young in balance posture?

9)The discussion of this paper should be modified.

Author Response

Please confirm the uploaded file.

Round 2

Reviewer 1 Report

Thank you for addressing my comments.

Author Response

Dear Reviewer 1,

Thank you for your comments.

It would be great if you could let us know which part should be modified in detail.

Sincerely yours,

Yusuke Maeda

Reviewer 2 Report

Thank you for your hard work, but there are some research design drawbacks in this paper can not be solved. The procedure of the test was not  clear. The discussion of this paper should be modified.

Author Response

Dear Reviewer 2,

We appreciate your crucial comments.

We modified our manuscript so please confirm uploaded file.

Sincerely yours,

Yusuke Maeda
